# Molecular Responses Mechanism of *Synechocystis* sp. PCC 6803 to Cadmium Stress

Gang Ruan [1,2], Wujuan Mi [1,*], Xuwang Yin [2], Gaofei Song [1] and Yonghong Bi [1,*]

[1] State Key Laboratory of Fresh Water Ecology and Biotechnology, Institute of Hydrobiology, Chinese Academy of Sciences, Wuhan 430072, China
[2] College of Fisheries and Life Science, Dalian Ocean University, Dalian 116000, China
* Correspondence: miwj@ihb.ac.cn (W.M.); biyh@ihb.ac.cn (Y.B.); Tel.: +86-27-68780016 (Y.B.)

**Abstract:** Cadmium is one of the major heavy metal pollutants in the environment. However, the toxicity targets and response mechanisms in photosynthetic organisms to cadmium are lacking. This study explored the physiological response of *Synechocystis* sp. PCC 6803 to cadmium stress; the toxicity targets and the resistance mechanism were screened. The results showed that cadmium led to an increase in reactive oxygen species content, superoxide dismutase activity, and the lipid peroxidation level, which damaged the cell's photosynthesis and ultrastructure. The cross-omics analysis found 1073 differentially expressed genes (DEGs), of which only 84 genes had the same expression trend at the mRNA and protein levels. The bioinformatics analysis found that the toxic effects of cadmium were inhibiting the expression of the proteins for the photosynthesis-antenna, photosynthesis, and ribosome pathway. The cell's response included the upregulation of proteins related to the two-component system, biosynthesis, and ABC transporter pathway. The study confirmed that the target sites of cadmium were chlorophyll *a* synthesis, photosynthetic electron transport chains, and ribosomes; the response mechanism to cadmium toxicity was the upregulation of the ABC transporter pathway and its protein expression. This study provides evidence to obtain insight into the toxicity targets and molecular response mechanism of *Synechocystis* sp. PCC 6803 to cadmium stress.

**Keywords:** heavy metal; microalgae; photosynthesis; transcriptome; proteome; molecular response

## 1. Introduction

Cadmium in the environment affects ecological security through biomagnification in the food chain [1,2], ultimately threatening human health. Previous studies demonstrated that cadmium was a carcinogenic, mutagenic, and endocrine-disrupting element [3]. Cadmium can cause lung damage and fragile bones or affect calcium regulation in the physiological system [4,5]. At present, cadmium has been officially registered as one of the top ten "major health problem chemicals" by the World Health Organization [6]. Studies on the effects of cadmium on higher plants have confirmed that cadmium causes cell damage and interferes with the homeostasis of copper, iron, manganese, and zinc [7,8]. Cadmium uses its chemical properties similar to those essential metal elements to enter the bacterial cell through protein channels of the cell membrane, such as metal importers of magnesium, manganese, and zinc, disrupting the intracellular ion balance [9].

Studies revealed that eukaryotic algae could tolerate and adapt to cadmium with low concentrations [5], and high concentrations of cadmium inhibited algal growth and damaged cells' ultrastructure, leading to reactive oxygen species (ROS) accumulation and activation of the antioxidant enzyme system [10]. The maximum photochemical efficiency was significantly reduced, and chloroplasts might be one of the targets of cadmium [11]. The metabolic pathways of photosynthesis, oxidative photosynthesis, glycolysis, TCA cycle, and ribosomal proteins biosynthesis of *Chlorella sorokiniana* were significantly downregulated under cadmium stress [5]. On the other hand, the gene expression of proteins or

transporters involved in absorption, sequestration, and detoxification also responds to cadmium toxicity [12,13]. Among them, transporters played a significant role in the resistance to cadmium stress, it was found in *Phaeodactylum tricornutum* that resistance mechanism might inhibit the increase of intracellular cadmium content by downregulation of influx transporter ZIP protein and upregulation of efflux transporter ATPase5-1B protein [14]. The critical elements of cadmium resistance in *Saccharomyces cerevisiae* and *Chlamydomonas reinhardtii* were ion transporter and ABC transporter proteins [15]. In addition, algal cells alleviated cadmium stress by adjusting the metabolism of macromolecular substances, such as glutamate, trehalose organic ligands, or extracellular polymeric substances (EPSs), which could effectively prevent the uptake of heavy metals by cells [16].

Cyanobacteria, as prokaryotic microalgae, have smaller genomes and are more sensitive to environmental changes than eukaryotic algae [6,17]. They grow fast, have no chloroplasts, and primarily use phycobilisome (PBS; the makeup of such as allophycocyanin (APC) and phycocyanin (PC)) instead of light-harvesting chlorophyll complexes to harvest light from the environment [18]. These differences lead to significant dissimilarities in the reaction/protective mechanism to cadmium between cyanobacteria and eukaryotic algae. Moreover, cyanobacteria have more promise for enriching cadmium in the aquatic environment [19,20]. Therefore, using cyanobacteria to study the toxicity and resistance mechanism of algae exposed to cadmium can obtain more interesting information. However, so far, there are very few studies on the responses of cyanobacterial cells to cadmium stress.

On the other hand, studies have found that cells have multiple defense pathways against cadmium stress: overexpression of bacterial metallothionein (*Bmt*A) chelating heavy metals is one of the vital defense mechanisms of cells [21]. *Smt*A is a typical *Bmt*A in *Synechococcus* PCC 7942, which contains four cysteines and two histidine and can chelate metal ions [22]. Besides these, metal efflux systems, including the multidrug resistance ABC transporter family (MDR channels) and the P-type ATP family involved in transporting metals such as zinc or copper, were a vital defense pathway for cadmium [7]. P-type ATPase *Znt*A was found to mediate cadmium resistance, and *Znt*A was closely related to the tolerance of cadmium [23]. The homologous protein of *Ars*R/*Smt*b, *azt*A/*azt*R was found in *Anabaena* PCC 7120 to function as a metal efflux pump [6], but whether the ABC transporter of algal cells plays a role in transporting heavy metals needs further research. Furthermore, the characteristics and mechanisms of the two-component system for sensing conditions changes and regulating gene expression in response to cadmium stress are still unclear [24]. It is necessary to perform further research on the response mechanisms of cyanobacteria to cadmium.

In order to gain insight into the response characteristics of cyanobacteria to cadmium, this study selected *Synechocystis* sp. PCC 6803 as the object. Algal cells were cultured under the cadmium concentration gradient (0.05, 0.25, 0.50, and 1.00 mg L$^{-1}$) for 72 h. The physiological activity, ultrastructure, transcription, and translation were screened to determine the physiological toxicity effect of cadmium; RNA-seq and Tandem Mass Tags (TMTs)-based quantitative proteomics were used to obtain differentially expressed genes/proteins to reveal the molecular mechanism of cadmium toxicity to cyanobacteria. This study contributed to an in-depth understanding of the cytotoxic effect of cadmium on algae and the response mechanism of algae cells to cadmium.

## 2. Materials and Methods

### 2.1. Algae Cultivation

*Synechocystis* sp. PCC 6803 was obtained from the Freshwater Algae Culture Collection at the Institute of Hydrobiology (FACHB-Collection, Wuhan, China). The strain was precultivated by using a sterile BG11 medium at $30 \pm 1$ °C, under continuous fluorescent white light at an intensity of 40 μmol photons m$^{-2}$ s$^{-1}$ and 130 rpm shaker incubation. The absorbance values of the cultures at 730 nm were measured by using a spectrophotometer

(UV1701, Shimadzu, Japan), and their growth was monitored by $OD_{730\,nm}$. Algal cells in the logarithmic growth phase ($A_{730\,nm}$ = 0.8–1.0) were collected for this study.

A stock solution of $CdCl_2 \cdot 2.5H_2O$ (China National Pharmaceutical Group Corporation) at 1.0 g $L^{-1}$ was diluted to the different test concentrations to add into the BG11 medium to gain the following final concentrations: 0 (control), 0.05, 0.25, 0.50, and 1.00 mg $L^{-1}$. Each treatment had three biological replicates. Algal cells with an initial inoculation concentration of $OD_{730\,nm}$ = 0.1 were added into an Erlenmeyer flask containing 100 mL BG11 culture solution with different cadmium concentrations. A sample with 1 mL was taken from each flask daily to monitor cell growth and measure $OD_{730\,nm}$ with a spectrophotometer [25]. Before use, all culture glassware was soaked overnight in 10% (*v/v*) hydrochloric acid, then washed with deionized and ultrapure water, and ultimately dried in a desiccator at 55 °C.

### 2.2. Measurement of Cadmium Concentrations

Cadmium was used at different concentrations to treat cells in the logarithmic growth phase for 72 h. Cells were collected by centrifugation (6000 *g*, 5 min) and washed three times, using ultrapure water. Afterward, the $OD_{730\,nm}$ of the culture was measured, and 0.35 mL algal samples were added to 2 mL of nitric acid, according to the microwave nitration, until clarified. A 7500a ICP–MS (Agilent Technologies, Tokyo, Japan) equipped with a Babington nebulizer was used to determine cadmium. [26]. The operating parameters for ICP–MS detection are listed in Supplementary Table S2.

### 2.3. Transmission Electron Microscopy (TEM) Assay

After 24 h and 72 h of treatment, the cells of *Synechocystis* sp. PCC 6803 were collected via centrifugation of 10 mL culture solution at 3000 *g* for 10 min and then fixed with 2.5% glutaraldehyde at 4 °C for 12 h. Dehydration of fixed cells and preparation of ultrafine sections were performed according to Ozaki [27]. The ultrastructure of cells was observed by using a Hitachi 7700 transmission electron microscope (Tokyo, Japan).

### 2.4. Biochemical Index Assays

The ROS content measurement: 1 mL of culture was incubated in 10 mM DCFH-DA for 1 h at 37 °C in the dark, with gentle shaking. Then the samples were washed three times by centrifugation with ultrapure water to remove the attached dye. Finally, 200 μL of culture was added to a 96-well plate. DCF fluorescence (excitation at 485 nm and emission at 530 nm) and chlorophyll fluorescence (excitation at 430 nm and emission at 670 nm) were detected by using a multi-mode enzyme-labeling Instrument (PerkinElmer VICTOR Nivo, Helsinki, Finland) to calculate ROS content [28].

Protein content, SOD activity, and malondialdehyde (MDA) content were measured by using Regent kits (Nanjing Jiancheng Biotechnology Institute, Nanjing, China). After 24 h and 72 h of exposure, we collected cells by centrifugation (4000 *g*) at 4 °C for 10 min. The pellets were homogenized in 1 mL PBS (pH = 7.2), using zirconium beads. After centrifugation (1800 *g*) for 10 min, the supernatant was used to determine the protein amount, SOD activity, and MDA content according to the regent kit manufacturer's instructions by measuring absorbance at 562 nm, 450 nm, and 530 nm, respectively, with a multi-mode enzyme-labeling Instrument (PerkinElmer VICTOR Nivo, Helsinki, Finland). We normalized the obtained value with the cell number or protein concentration to calculate the final value.

### 2.5. Measurement of Pigment Content, Chlorophyll Fluorescence, and $Q_A^-$ Reoxidation Kinetics

Chlorophyll *a* (Chl *a*) content measurement: Samples with 5 mL were centrifuged 4000 rpm for 10 min. The pellets were resuspended in 90% methanol and left overnight at 4 °C in the dark, and then they were centrifuged again. The absorbance of the supernatant was measured by using a spectrophotometer (UV1701, Shimadzu, Japan) at wavelengths of 665 and 652 nm, and the concentrations of pigments were calculated by using the following equations: chlorophyll *a* (mg $L^{-1}$) = 16.82 $A_{665}$ − 9.28 $A_{652}$ [29,30].

Values of $Fv/Fm$ (The maximum quantum yields of photosystem II(PSII)) were determined with an AquaPen-C AP-C 100 instrument (Photon Systems Instruments, Drasov, Czech Republic); algal cells, after 15 min of dark adaptation, were used to test chlorophyll fluorescence and obtain this value [31,32].

$Q_A^-$ reoxidation kinetics of algal cells treated with different cadmium concentrations for 72 h were measured with an FL-6000 dual-modulation kinetic fluorometer (Photon Systems Instruments, Brno, Czech Republic) [33,34]. The $Q_A^-$ reoxidation kinetics was fitted by a three-component exponential equation for quantitative evaluation:

$$F(t) = A1 \, exp \, (-t \, / \, T1) + A2 \, exp \, (-t \, / \, T2) + A3 \, exp \, (-t \, / \, T3) + A0$$

where $F(t)$ is the fluorescence yield at time $t$, $A0$ to $A3$ are the amplitudes, and $T1$ to $T3$ are the time constants from which the half-life value could be calculated as $t_{1/2} = \ln 2 \, T$.

*2.6. Combined Analysis of Transcriptome and Proteome*

Algal cells were treated with a cadmium concentration of 0.50 mg $L^{-1}$ for 72 h, taking 50 mL of algae solution, centrifuging (4000 $g$, 4° C, 10 min) to collect algal cells, and grinding and crushing with liquid nitrogen. Total RNA was extracted from the tissue by using TRIzol® Reagent according to the manufacturer's instructions (Invitrogen, Waltham, MA, USA), and genomic DNA was removed by using DNase I (Takara, Japan). Then RNA quality was determined by using 2100 Bioanalyser (Agilent, Santa Clara, CA, USA) and quantified by using the ND-2000 (NanoDrop Technologies, Wilmington, NC, USA). A high-quality RNA sample (OD260/280 = 1.8~2.2, OD260/230 ≥ 2.0, RIN ≥ 6.5, 28S:18S ≥ 1.0, >10 μg) was used to construct the sequencing library. RNA-seq strand-specific libraries were prepared by following the TruSeq RNA sample preparation Kit from Illumina (San Diego, CA, USA), using 5 μg of total RNA. Shortly, rRNA was removed by a RiboZero rRNA removal kit (Epicenter) and fragmented by using fragmentation buffer. Then cDNA synthesis, end repair, A-base addition, and ligation of the Illumina-indexed adaptors were performed according to Illumina's protocol. Libraries were then sized for 200–300 bp cDNA target fragments on 2% Low Range Ultra Agarose, followed by PCR amplification, using Phusion DNA polymerase (NEB) for 15 PCR cycles. After being quantified by TBS380, paired-end libraries were sequenced by Illumina NovaSeq 6000 sequencing. In Linux, the FastQC package evaluated the quality of the raw reads. The raw paired-end reads were trimmed and controlled by Trimmomatic. Then clean reads were separately aligned with the reference genome with orientation mode, using the Rockhopper package. The gene expression levels were normalized by using the fragments per kilo bases per million (FPKMs) fragments method. In the R (version 4.0.5) software, the DESeq2 package was used to determine the differential expression of the analysis [35–37]. Genes with a log2 (fold change) of ≥1 and ≤−1 compared with the control group at the adjusted $p$-adjust of <0.05 (according to false discovery rate (FDR) correction) were identified as differentially expressed genes (DEGs).

The proteins were extracted with SDT lysis buffer and labeled with the Tandem Mass Tag (TMT) reagent 8-plex kit (Thermofisher, Waltham, MA, USA) following the protocol manuals. The Cadmium treatment group was labeled with TMT tags 128 C, 129 N, and 129 C; the control group was labeled with 126, 127 N, and 127 C. The peptides were fractionated by high-pH reversed-phase chromatography, using an Agilent 1260 Infinity HPLC system (Agilent, CA, USA). After that, LC–MS/MS data acquisition was performed on a Q Exactive HF-X mass spectrometer (Thermo Fisher Scientific, Waltham, MA, USA) and a nanoEasy-nLC 1200 system (Thermo Fisher Scientific, Waltham, MA, USA). Experimental procedures refer to previous studies [38,39]. Proteins were identified by using PROTEOME DISCOVERER (Thermo Fisher Scientific, Waltham, MA, USA) according to the UNIPROT database of *Synechocystis* sp. PCC 6803 (https://www.uniprot.org/proteomes/UP000001425) (accessed on 18 March 2022). Proteins with a fold change >1.50 or <0.67 were identified as differentially expressed proteins (DEPs) based on t-tests at $p < 0.05$ [40].

### 2.7. Statistical Analysis

SPSS (Version 25.0, SPSS Inc., Chicago, IL, USA) software and two-way analysis of variance (ANOVA), followed by a least significant difference (LSD) post hoc test, were used to check the significant difference between treatments.

According to the alignment of the Gene Ontology (GO) and (KEGG) database, the cluster and enrichment analysis of DEGs/DEPs was completed in R (version 4.1.0), using the Tidyverse package and the ClusterProfiler package [41]. According to the STRING 11.5 database, a protein–protein interaction (PPI) network and Molecular Complex Detection Algorithm (MCODE) containing differentially expressed proteins with significantly enriched ($p < 0.05$) functional modules were constructed by using Cytoscape (version 3.8.0) software [42].

## 3. Results

### 3.1. Intracellular Cadmium Content and Growth Inhibition

Compared with the control group, cadmium was accumulated significantly in cells treated with a cadmium concentration higher than 0.25 mg $L^{-1}$ (Figure 1A), and cell growth was significantly inhibited. In particular, the maximum growth inhibition rates were 26.42%, 69.42%, and 85.98% after 72 h exposure at 0.25, 0.50, and 1.00 mg $L^{-1}$ (Figure 1B), respectively, and the median effect concentration caused 50% growth inhibition ($EC_{50}$), which was close to 0.50 mg $L^{-1}$.

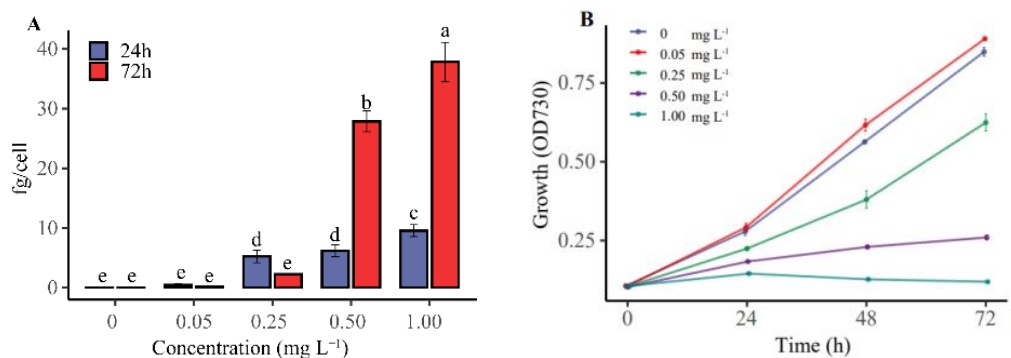

**Figure 1.** Algal cells were treated with cadmium at different concentrations for 72 h. (**A**) Intracellular cadmium content. (**B**) Growth curve. The mean and standard deviation of three replicates were shown in the figure. According to the one−way ANOVA test, different alphabet letters indicated a significant difference in different groups at $p < 0.05$.

### 3.2. TEM Analysis

The concentrations of cells exposed to 0.50 mg $L^{-1}$ cadmium for 24 h and 72 h are shown in the Figure. 2. An intact cell membrane structure, uniform thylakoid lamellae, and contiguous EPS were observed in normal cells (Figure 2A). After treatment with 0.50 mg $L^{-1}$, the thylakoid lamellae were broken, and cell membrane collapse and EPS dispersion were observed (Figure 2B,C). Furthermore, the damage was more evident after 72 h than 24 h.

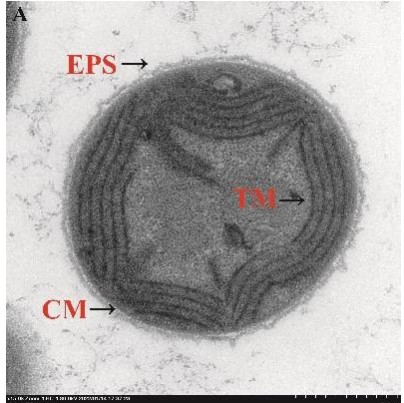
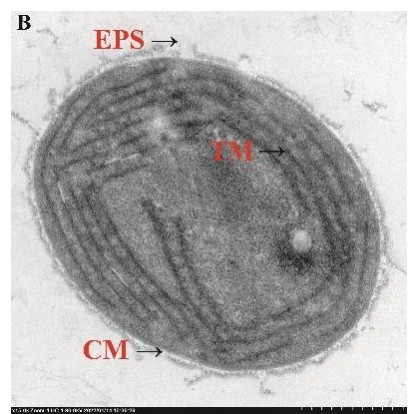
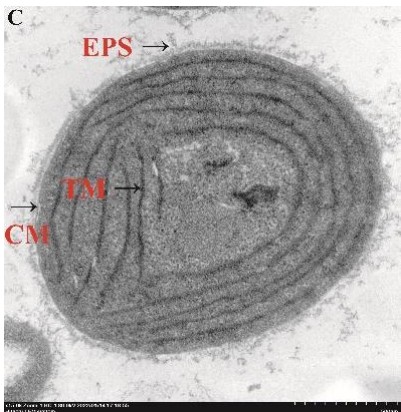

**Figure 2.** TEM image of algal cells treated with 0.50 mg L$^{-1}$ cadmium. Thylakoid membrane (TM), cell membrane (CM), and extracellular polymeric substances (EPSs). (**A**) Control group, (**B**) treatment for 24 h, and (**C**) treatment for 72 h.

*3.3. ROS Content, SOD Activity, and MDA Content*

When the cadmium concentration was higher than 0.25 mg L$^{-1}$, the ROS content, SOD activity, and MDA content increased significantly ($p < 0.05$) compared with the control. After 72 h of exposure, the ROS content increased by 128.66%, 452.38%, and 404.42%, respectively; the SOD activity increased by 29.87%, 338.36%, and 647.75%, respectively; and the MDA content increased significantly by 305.74%, 375.82%, and 242.74%, respectively (Figure 3).

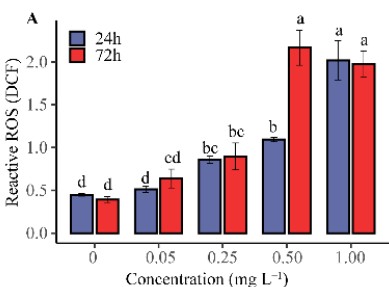
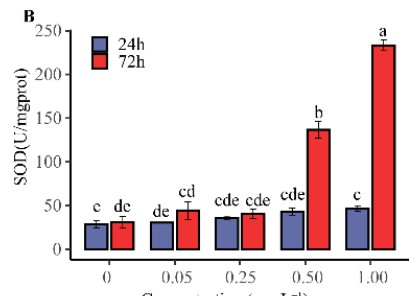
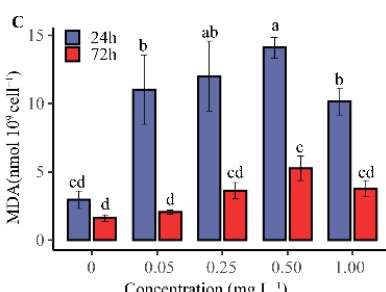

**Figure 3.** Algal cells were treated with cadmium at different concentrations for 24 h and 72 h. (**A**) ROS content, (**B**) SOD activity, and (**C**) MDA content. According to the one-way ANOVA test, different alphabet letters indicated a significant difference in the treatment group at $p < 0.05$.

*3.4. Pigment Content and Photosynthetic Activity*

No significant changes in chlorophyll *a* content ($p > 0.05$) were observed at 0.05 mg L$^{-1}$ cadmium exposure. At the same time, it significantly decreased above 0.25 mg L$^{-1}$ ($p < 0.05$) compared to the control group (Figure 4A). Furthermore, the results showed that the *Fv/Fm* values of algal cells were significantly decreased ($p < 0.05$) under cadmium exposure above 0.25 mg L$^{-1}$ compared to the control group (Figure 4B).

To further demonstrate the cadmium-induced inhibition of the acceptor side of PSII, the $Q_A^-$ reoxidation kinetics were measured (Figure 4C), and the fluorescence curves were further explored by fitting them to a three-component exponential decay. The results showed that $Q_A^-$ reoxidation kinetics were not affected by 0.05 mg L$^{-1}$ cadmium treatment. When the cadmium concentration was increased to 0.50 mg L$^{-1}$, the time constants of the fast phase of the control cells increased from 593 µs to 670 µs, respectively. Meanwhile, the time constants of the middle phase increased from 7.76 ms to 8.40 ms, respectively, during this period. The fast and middle phases increased further when the cadmium concentration was further increased (Table 1).

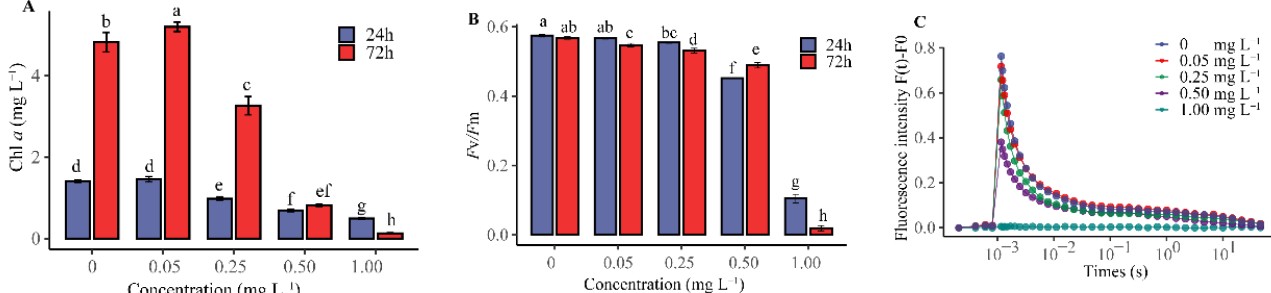

**Figure 4.** Algal cells were treated with cadmium at different concentrations for 24 h and 72 h. (**A**) Chlorophyll *a* content. (**B**) *Fv*/*F*m. According to the one-way ANOVA test, different alphabet letters indicate a significant difference in different treatment groups at $p < 0.05$. (**C**) Treatment time of $Q_A^-$ reoxidation kinetics of different concentrations of cadmium was 72 h.

**Table 1.** Decay kinetics of flash-induced variable fluorescence in *Synechocystis* sp. PCC 6803 was treated with different cadmium concentrations for 72 h. The curves were analyzed by using three exponential components (fast, middle, and slow phases).

| Concentration /(mg L$^{-1}$) | Fast Phase | | Middle Phase | | Slow Phase | |
|---|---|---|---|---|---|---|
| | A1/(%) | T1/(µs) | A2/(%) | T2/(ms) | A3/(%) | T3/(s) |
| 0 | 91.13 | 593 | 6.75 | 7.76 | 2.12 | 15.29 |
| 0.05 | 91.30 | 575 | 6.35 | 8.88 | 2.35 | 15.27 |
| 0.25 | 94.91 | 421 | 4.14 | 4.97 | 0.96 | 17.92 |
| 0.50 | 87.23 | 670 | 8.14 | 8.40 | 4.63 | 4.22 |

### 3.5. Differentially Expressed Genes

The transcriptome analysis showed that more than 29% of the genes showed significant differential expression in response to 0.50 mg L$^{-1}$ cadmium stress compared with the control. Among them, 527 genes were upregulated, and 546 were downregulated (Figure 5A and Supplementary Table S3). Furthermore, the hierarchical clustering of DEGs showed that genes with similar functions or involved in identical biological processes were clustered in the same group. Good reproducibility of DEGs expression was observed in samples within the same group.

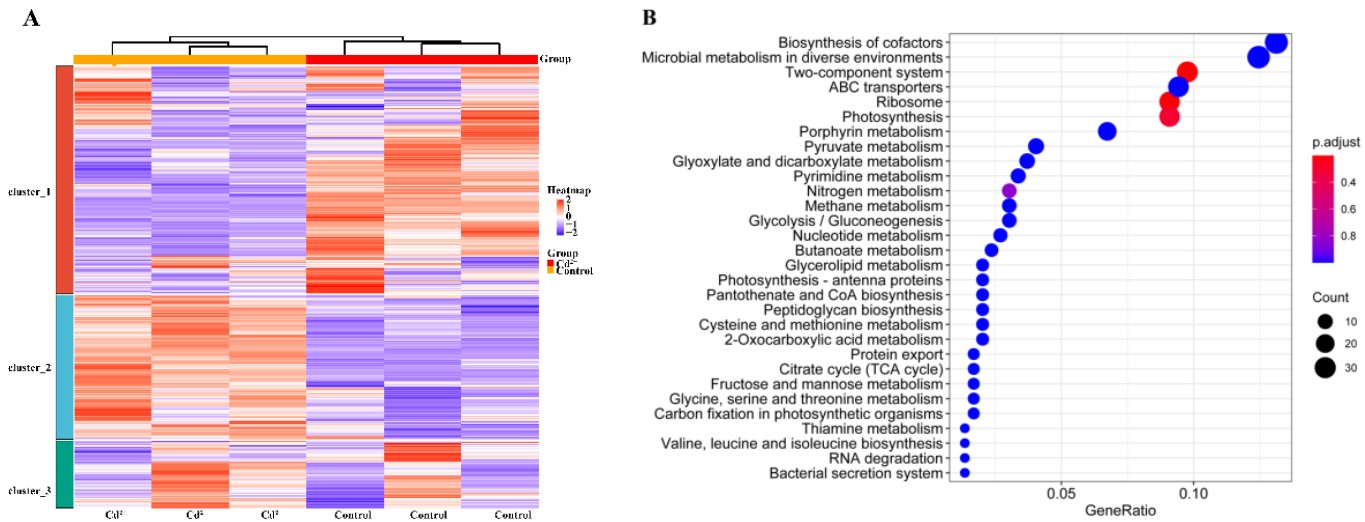

**Figure 5.** (**A**) Heat map of abundance distribution expressed by DEGs. (**B**) Dot plots of DEGs enrichment analysis by KEGG.

To determine the functional categories of DEGs, statistically significant differentially expressed genes were subjected to an enrichment analysis based on the KEGG database, which showed 581 DEGs clustered into 64 KEGG pathways (Figure 5B and Supplementary Table S4). Among these, genes enriched in the metabolism process accounted for the largest proportion (219), followed by genes enriched in the biosynthesis of cofactors (39), two-component system (29), ABC transporters (28), ribosome (27), photosynthesis (27), biosynthesis of amino acids (18), oxidative phosphorylation (10), Glycolysis/Gluconeogenesis (9), biosynthesis of nucleotide sugars (7), aminoacyl-tRNA biosynthesis (7), photosynthesis-antenna proteins (6), pantothenate and CoA biosynthesis (6), peptidoglycan biosynthesis (6), protein export (5), citrate cycle (TCA cycle) (5), carbon fixation in photosynthetic organisms (5), and quorum sensing (5). Three pathways were significantly enriched in DEGs ($p < 0.05$), respectively, two-component system (29), ribosome (27), and photosynthesis (27).

### 3.6. Differentially Expressed Proteins

The proteome analysis showed that more than 13% of the proteins exhibited significant differential expression in response to cadmium exposure with 0.50 mg L$^{-1}$ compared to the control group. Among them, 148 proteins were upregulated, and 157 were downregulated (Supplementary Table S5). A total of 272 of the 305 differentially expressed proteins were involved in 681 pairwise interactions, and a PPI network containing significantly enriched ($p < 0.05$) functional modules was constructed (Figure 6B). The hub proteins in the PPI network were imputed by MCODE to be associated with the translation process. They were shown to be downregulated (Figure 6C and Supplementary Table S7).

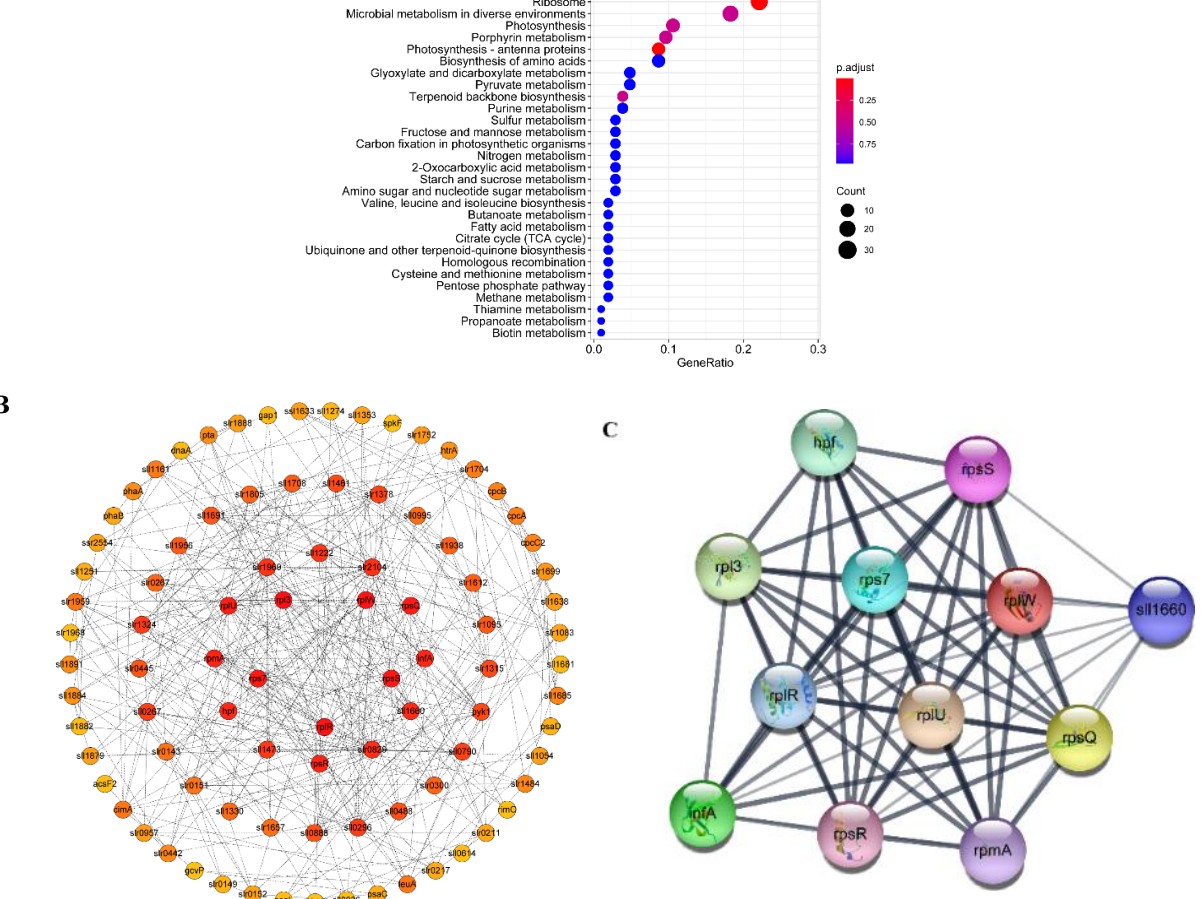

**Figure 6.** (**A**) Dot plots of DEPs enrichment analysis by KEGG. (**B**) The interaction network of the DEPs containing. (**C**) Top Protein–protein interaction networks in MCODE analysis.

To determine the functional categories of DEPs, an enrichment analysis was performed by using the KEGG database, which showed 210 DEPs clustered into 46 KEGG pathways (Figure 6A and Supplementary Table S6). The maximum number of proteins (81) were enriched in the metabolism process, accounting for 38.57% of the total number of enriched proteins, followed by the biosynthesis of cofactors. Similarly, the protein numbers of microbial metabolism in diverse environments, porphyrin metabolism, carbon metabolism, pyruvate metabolism, and pyruvate metabolism in the metabolic pathway were highly enriched with 19, 10, 6, 5, and 5 proteins, respectively. Twenty-two metabolism-related pathways were identified from KEGG analysis. Other highly enriched pathways included ribosome (29), photosynthesis (11), biosynthesis of cofactors (11), biosynthesis of amino acids (9), photosynthesis-antenna proteins (8), terpenoid backbone biosynthesis (4), two-component system (4), carbon fixation in photosynthetic organisms (3), oxidative phosphorylation (3), and ABC transporters (3). In total, four pathways were significantly enriched in DEPs ($p < 0.05$), respectively, ribosome (29), porphyrin metabolism (10), and photosynthesis-antenna proteins (8), terpenoid backbone biosynthesis (4).

A low correlation between transcriptome and proteome expression was observed in *Synechocystis* sp. PCC 6803 exposed to cadmium (Figure 7A). The expression abundance of 60 upregulated genes and 24 downregulated genes was consistent with the abundance of their expressed proteins. In comparison, 49 genes and their expressed proteins showed opposite expression patterns in the proteome and transcriptome, respectively (Figure 7B). The 84 genes or proteins with consistent expression were subjected to GO clustering analysis, and the downregulated genes/proteins were clustered into photosynthesis (*psaD*, *cpcC2*, *cpcA*, *psbB*, and *psbA2*), translation (*rplR*, *rpsQ*, *rpsS*, *rplW*, *rplC*, *hpf*, and *rpoD*), and motility (*slr2015*, *hofG*). Moreover, the upregulated proteins were clustered into signal transduction (*ctpB*, *prc*, *sll1708*, *sphS*, and *sll0537*), oxidoreductase (*slr1608*, *fabG*), protein maturation (*hslO*, *lipA1*, and *htrA*), biosynthetic process (*degT*, *ggpS*, *alr*, *thiC*, *sps*, *rfbM*, *slr0897*, *slr0955*, and *leuA*), membrane biosynthetic process (*murI* and *sll1377*), enzyme activity (*sll1723* and *gloA*), transport (*pstB3*, *zntC*, *sll1725*, *slr0944*, *arsH*, *sll0142*, and *slr0544*), metalloendopeptidase activity (*slr1708*), and DNA replication (*dnaA*) (Figure 8).

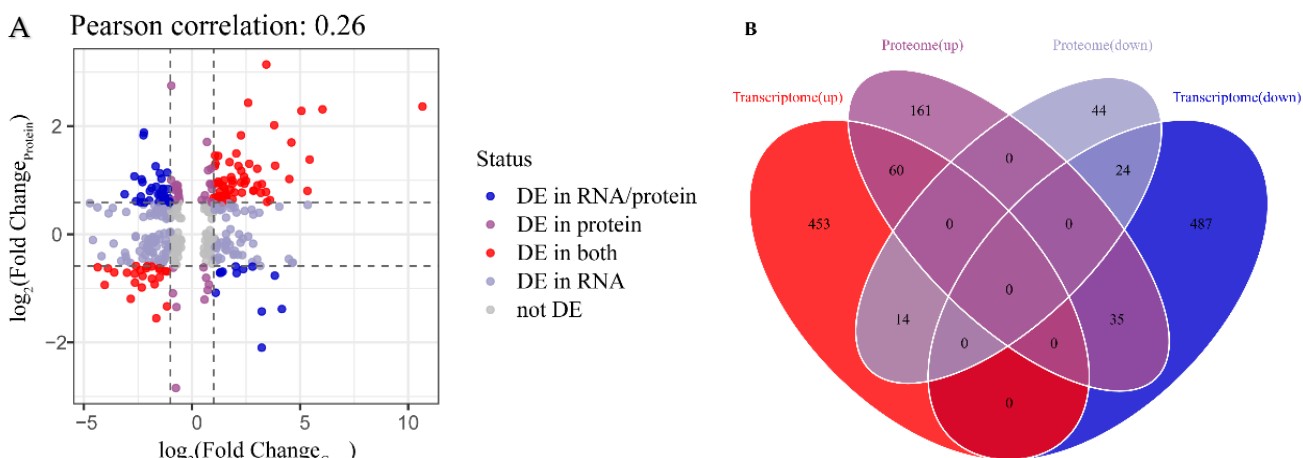

**Figure 7.** (**A**) Scatter diagram illustrating the relationship between differentially expressed genes and proteins quantified in transcriptomic and proteomic data sets. DE, differential expression. (**B**) Venn diagram showing the overlap of the differentially expressed genes and proteins.

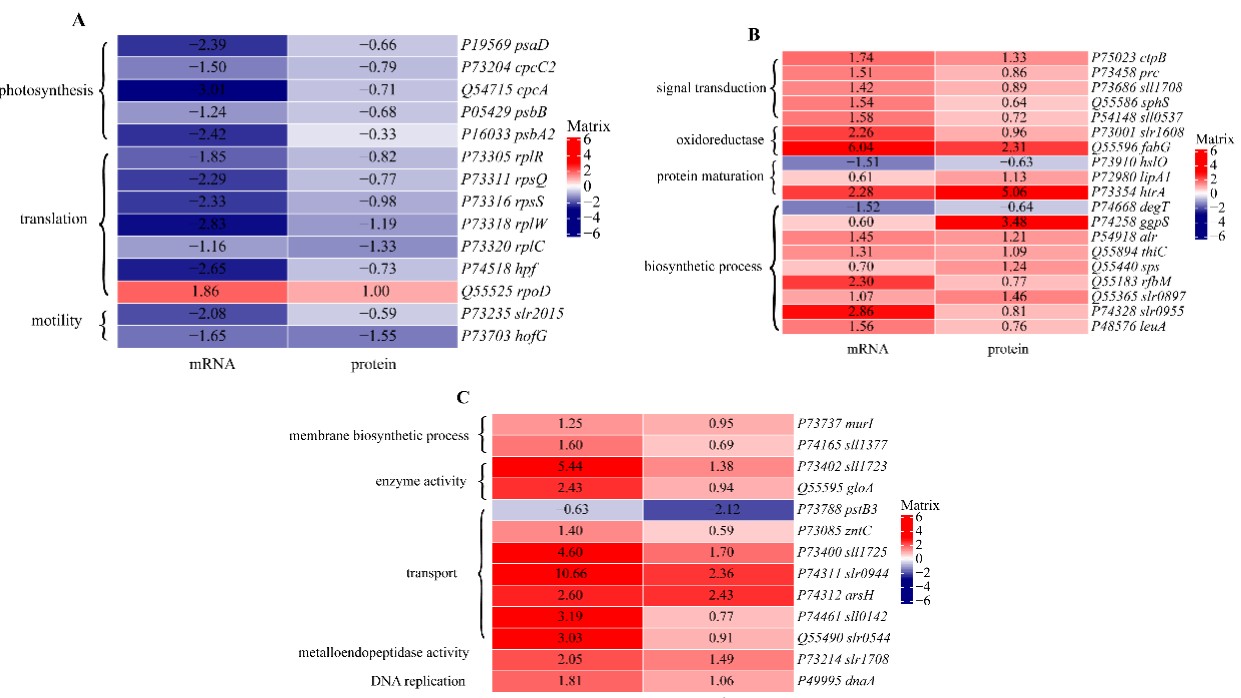

**Figure 8.** Heat map of differentially expressed genes and protein abundance with consistent expression. (**A**) GO clustering in photosynthesis, translation, and mobility. (**B**) GO clustering in signal transduction, oxidoreductase, protein maturation, and biosynthetic process. (**C**) GO clustering in membrane biosynthetic process, enzyme activity, transport, metalloendopeptidase activity, and DNA replication.

## 4. Discussion

### 4.1. Effects of Cadmium on the Physiological Activity

*Synechocystis* sp. PCC 6803 is sensitive to cadmium stress because of its uncomplicated structure. Our study demonstrated that the growth of *Synechocystis* sp. PCC 6803 was significantly inhibited under cadmium's concentration higher than 0.25 mg L$^{-1}$. Similarly, previous studies found that the growth of *Microcystis aeruginosa* was inhibited, and the photosynthetic system was damaged due to intracellular enrichment of high doses of cadmium [43]. When the cadmium concentration in the culture system was higher than the threshold, cadmium was considered to enter the cell through cation transport systems on the plasma membrane. These transporters mainly comprise *p*-type ATPases, ZIP, and Natural Resistance-Associated Macrophage Proteins (NRAMPs) family or Ca$^{2+}$ channels [44]. Cadmium ions entering cells can disrupt the balance of intracellular ROS generation and scavenging. Trace elements play pivotal roles in cyanobacteria's photosynthetic electron transport chain, making it susceptible to direct interference by cadmium. The oxygen-rich thylakoid lamellae are accompanied by a strong electron flow overflow of the damaged electron transfer chain, which is considered the primary way to generate ROS [45]. Therefore, photosynthetic-system damage may be the ROS accumulation. The accumulation of ROS stimulates the intracellular antioxidant system, leading to an increase in the activity of antioxidant enzymes such as SOD. When the antioxidant system fails to scavenge excess ROS, ROS further accumulates, and the cells suffer from oxidative damage and lipid peroxidation, leading to increased MDA content and impaired membrane lipid structure [46]. The ultrastructure of the cells showed disruption of the thylakoid lamellae and cell membrane structure, further confirming the cadmium-induced oxidative damage, with results similar to other studies in which heavy metals affected the ultrastructure of microalgae [47,48].

In previous studies, cadmium was considered to affect photosynthetic efficiency seriously [49,50]. In the same way, cadmium above 0.25 mg L$^{-1}$ significantly reduced pho-

tosynthetic efficiency in this study. Reduced photosynthetic efficiency results in insufficient photosynthetic products to support their growth needs. Chlorophyll *a* is an indispensable macromolecule in photosynthesis. The results showed that cadmium above 0.50 mg L$^{-1}$ significantly inhibited the synthesis of chlorophyll *a*. A fitting analysis of the Q$_A$$^-$ redox kinetic curve is commonly used to understand the effect of stress on the PSII electron transport chain. Our result showed that cadmium treatment for 72 h at concentrations higher than 0.50 mg L$^{-1}$ significantly prolongs the duration of fast and middle phases and that cadmium affects the electron transfer between plastoquinone (PQ), Q$_A$$^-$, and Q$_B$ [33,51]. Therefore, the inhibition of chlorophyll *a* synthesis and impaired PSII electron transfer were considered responsible for decreased photosynthetic efficiency.

### 4.2. Effects of Cadmium on Gene Transcription

Cyanobacteria perceive external stresses mainly through the signal transduction system, in which the two-component system plays a critical role in cyanobacteria's perception of environmental stresses and regulates cellular activities. In this study, we analyzed DEGs by KEGG enrichment. We found that 29 genes were significantly enriched to the two-component system at the transcriptome level (Figure 5B), demonstrating that the DEGs play a key role in response to cadmium stress and may regulate transcription and protein expression. Meanwhile, the result showed that 27 genes significantly enriched the ribosome pathway. Ribosomes play a pivotal role in cells' biological protein synthesis, growth, and development [52]. Ribosome transcriptional regulation produced a marked effect on protein metabolism. Transcript expression levels of ribosomal genes have been reported to be significantly downregulated by exposure to cadmium in *Dunaliella salina* [47]. Studies have shown that replacing cadmium with zinc, magnesium, manganese, and iron leads to changes in the biochemical properties of photosynthesis-related proteins. Cadmium entering cells is believed to directly affect photosynthesis efficiency [49]. Moreover, gene regulation at the transcriptional level leads to changes in the expression of related proteins, affecting photosynthetic system assembly. Our result showed that 27 genes in the photosynthesis pathway were significantly enriched, including the downregulation of PSII reaction center protein D1 (*psbA2* and *psbA1*) gene expression, which proved that cadmium could inhibit the expression of photosynthesis-related genes, thus affecting the efficiency of photosynthesis. Genes that regulate the expression of metal transporters in cells have received the most attention [53]. A cross-kingdom comparative transcriptomic analysis was conducted for *E. coli*, *S. cerevisiae*, and *C. reinhardtii* under cadmium stress and found that the key pathway of cadmium resistance is enriched in metal ion transport, and the key proteins may be ion transporter and ABC transporter [15]. *Anabaena* sp. was also enriched in the ABC transporters pathway during the recovery phase after cadmium exposure, and the enriched proteins were shown to be upregulated [54]. Our result showed that 28 genes were enriched in the ABC transporters pathway and that the transport system on the cell membrane was actively mobilized to reduce the intracellular cadmium concentration in the presence of cadmium, which is a key mechanism for the response to cadmium toxicity in *Synechocystis* sp. PCC 6803.

### 4.3. Effect of Cadmium on the Synthesis of Proteins

The analysis of DEPs by KEGG showed the response mechanism of cadmium stress in *Synechocystis* sp. PCC 6803 at the protein level firstly by regulating the efficient expression of proteins in the signal transduction system to sense extracellular cadmium stress, which affects transcription factors to regulate the translation of intracellular ribosomal subunits and controls the assembly of the photosynthetic system and ABC transporter. Among them, *spkF*, *slr1969*, *slr1363*, *slr0152*, *slr2104*, *slr1805*, *slr1324*, and *slr6001* belong to histidine/serine kinases whose protein expression was significantly affected by cadmium stress, triggering regulation of genes and proteins. The photosynthetic system pathway was enriched with proteins, including *psbB*, *petF*, *petC1*, *petB*, *psaL*, *psaD*, *psaC*, *psaF*, *atpG*, and *atpD*, which are the core components of four major protein complexes of the photosynthetic system

(PS II, ferredoxin, PS I, and ATP synthase). The cadmium stress prevents these proteins from assembling effectively, resulting in a significant decrease in photosynthetic efficiency. The harvesting light in cyanobacteria occurs mainly through phycobilisomes (PBSs) on the surface of the thylakoid lamella, which is mainly composed of many phycobiliproteins and linker polypeptides. Moreover, the phycobiliproteins comprising allophycocyanin (APC) and phycocyanin (PC) proteins are present in the photosynthesis-antenna proteins pathway [55]. The result showed that the *cpcA*, *cpcB*, *cpcC1*, *cpcC2*, *cpcL*, *apcC*, *apcD*, and *apcD* proteins were significantly enriched in the photosynthesis-antenna proteins pathway. It indicated that the synthesis of light-trapping antenna proteins in *Synechocystis* sp. PCC 6803 was affected by cadmium, resulting in inefficient absorption of light energy, which may be one of the main reasons for the decrease in photosynthetic efficiency. Furthermore, a similar finding was made in a previous study of the response of *Synechocystis* sp. PCC 6803 to titanium dioxide nanoparticles [22]. PPI shows the interaction between different proteins (Figure 6B) that is located in the inner ring and has a deep color, indicating that the protein is closely related to other proteins. Figure 6C shows that the hotspot proteins obtained by the MCODE algorithm are enriched in the ribosome pathway, thus further confirming that the synthesis of ribosomal subunits in *Synechocystis* sp. PCC 6803 is the main target of cadmium.

### 4.4. Molecular Response Mechanism of Synechocystis sp. PCC 6803 to Cadmium Stress

The bioinformatics analysis revealed 29% significantly differential genes at the transcriptome level but only 13% significantly differential proteins at the proteome level. Differentially expressed genes and proteins had a low correlation (0.12) (Figure 7A), showing an inconsistency between gene-level and protein-level expression. The difference between transcriptome and proteome expression may be due to different degradation rates between mRNA and protein [56]. Furthermore, studies in eukaryotes have suggested that structural changes in mRNA could cause such differences [57]. Similarly, some studies found a low correlation between cyanobacterial genes and proteins' expression under antibiotic or high light stress [58].

We selected differentially expressed genes and proteins that were overlapping and consistently expressed fractions, and we analyzed their response to cadmium by conducting a cross-omics comparison to obtain a more accurate model of the cadmium response in *Synechocystis* sp. PCC 6803. As a result, 84 proteins and corresponding genes were obtained (Figure 7B). The GO clustering showed that *psaD*, *cpcC2*, *cpcA*, *psbB*, and *psbA2* were significantly downregulated in the photosynthesis pathway. The inefficient assembly of these components results in a defective electron transport chain, in which electron transport between PQ, $Q_A^-$, and $Q_B$ is significantly slowed down, further confirming that these five proteins in the photosynthetic system responded significantly to cadmium. Meanwhile, *rplR*, *rplW*, *rplC*, *rpsQ*, *rpsS*, and *hpf* were significantly downregulated in the ribosome pathway, and they may be targets of the cadmium effects. Inhibition of the ribosomal subunit synthesis may slow down material and energy consumption during translation in cadmium-exposed *Synechocystis* sp. PCC 6803, and cells can divert limited materials and energy to critical pathways for cadmium resistance. However, the change of ribosome expression may lead to the disorder of the metabolic pathway of supporting cell growth, manifested by slowing down cell proliferation. The upregulated proteins were involved in nine GO biological processes, and cells were used to regulate downstream pathways to resist cadmium stress by upregulating the expression of signal transduction-related proteins. The biosynthetic process proteins, including *ggpS*, *alr*, *thiC*, *sps*, *rfbM*, *slr0897*, *slr0955*, and *leuA*, were efficiently expressed, and the upregulation of biomacromolecule synthesis may facilitate the cellular resistance to stress-induced cytotoxicity [59]. In addition, the secretion of the EPS composed of biomolecules to form EPS was considered the first line of defense for chelating heavy metals [60]. Meanwhile, *slr1608* and *fabG* clustering to redox activity at the proteome level further confirmed that cellular antioxidant activity was activated.

The membrane system of cyanobacteria is significantly affected by cadmium (Figure 2). The upregulated expression of the cell wall and membrane-synthesis-related proteins *murI* and *sll1377* in our study reflected a cellular resistance mechanism. They are the main physical barriers that restrict the entry of cadmium into cells. Their damage prevents cells from efficiently absorbing nutrients and limits cellular resistance to stress and growth [61]. Ion transporters on the cell membrane are essential components of microalgae resistance to heavy metal stress. Moreover, the inhibition of cadmium uptake and enhancement of cadmium efflux are regarded as the main mechanisms of cadmium resistance [62]. ABC transporters and ATPases of cyanobacteria may be considered in cadmium efflux [13]. Proteins significantly upregulated in association with the transporters were *zntC*, *sll1725*, *slr0944*, *arsH*, *sll0142*, and *slr0544*; they are considered cation efflux system proteins and ABC transporters, which play a crucial role in the resistance to cadmium stress. The above results suggest a possible response mechanism of cadmium of *Synechocystis* sp. PCC 6803 as suggested in Figure 9.

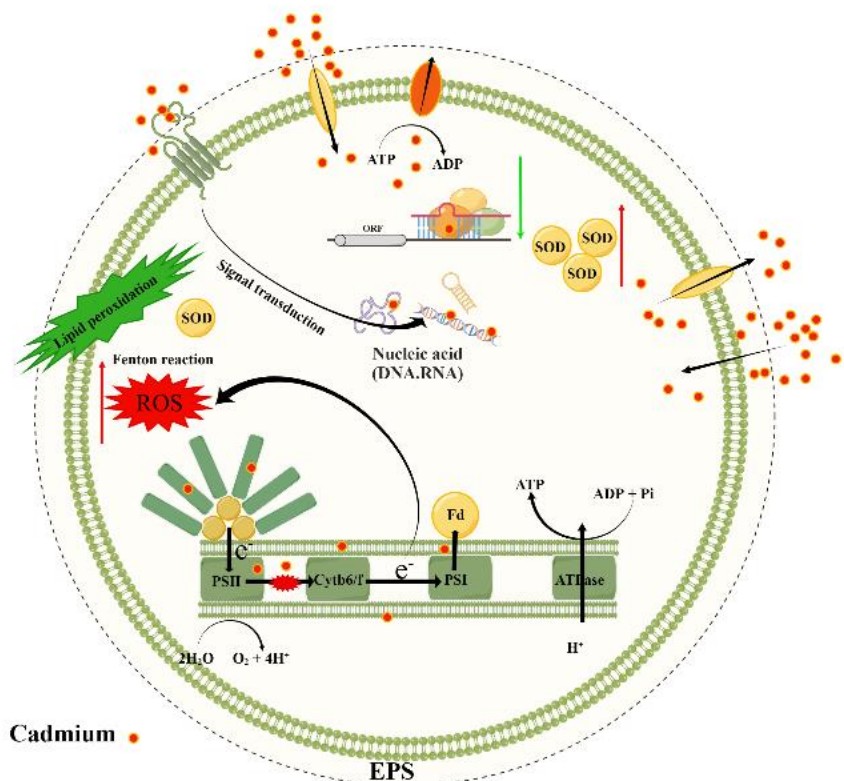

**Figure 9.** A conceptual model of the mechanism of cadmium action on *Synechocystis* sp. PCC 6803.

## 5. Conclusions

Damage to the photosynthetic system, resulting in ROS production, was caused by the massive entry of cadmium into the cells. Excessive ROS causes an increase in the activity of the antioxidant enzyme SOD, causing lipid peroxidation and destabilizing cell membranes and thylakoid lamellae. The results of the cross-omics analysis showed that cells respond to cadmium stress by showing a high expression of sensor proteins in the two-component system, followed by direct regulation of genes and proteins' expression of transcription and translation. It triggered the downregulation of the subunits' proteins in the photosynthetic system multiple protein complexes and ribosomal, resulting in slowing cell proliferation. Moreover, the overexpression of the ABC transporter pathway proteins on the cell membrane proved that ion efflux was the principal resistance mechanism of *Synechocystis* sp. PCC 6803 to cadmium stress.

**Supplementary Materials:** The following supporting information can be downloaded at https://www.mdpi.com/article/10.3390/w14244032/s1, Table S1: Culture condition of *Synechocystis* sp. PCC 6803; Table S2: Operating parameters of ICP-MS; Table S3: The identified results of differentially expressed genes in the cadmium treated group compared with the solvent control group; Table S4: Enrichment results of KEGG pathways for differentially expressed genes in cadmium treated group according to the KEGG database; Table S5: The identified results of differentially expressed proteins in the cadmium treated group compared with the solvent control group; Table S6: Enrichment results of KEGG pathways for differentially expressed proteins in cadmium treated group according to the KEGG database; Table S7: The top twelve most highly connective hub proteins in the PPI network of cadmium treated group

**Author Contributions:** Investigation, G.R. and W.M.; conceptualization, G.R. and Y.B.; data curation, G.R., W.M. and G.S.; methodology, G.R. and X.Y.; funding acquisition, Y.B.; supervision, Y.B.; writing—original draft preparation, G.R.; writing—review and editing, Y.B. All authors have read and agreed to the published version of the manuscript.

**Funding:** This work was jointly supported by National Key Research and Development Project (No. 2020YFA0907402; No. 2021YFC3200900) and National Natural Science Foundation of China (No. 31971477).

**Data Availability Statement:** The data presented in this study are available upon request from the corresponding author.

**Conflicts of Interest:** The authors declare no conflict of interest.

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
