# Peer review of "Molecular Responses Mechanism of Synechocystis sp. PCC 6803 to Cadmium Stress"

_water, doi:10.3390/w14244032_

Round 1

Reviewer 1 Report

The size of the figures and its lables in this manucript should be rearranged and aligned to reach a unified style. 

Reviewer 2 Report

Title: Molecular Responses Mechanism of Synechocystis sp. PCC 6803 to Cadmium Stress
Authors: Gang Ruan et al..

Cadmium is one of the major heavy metal pollutants but its toxicity targets on photosynthetic organisms are lacking. In this manuscript, the physiological response of Synechocystis sp. PCC 6803 to cadmium stress were explored and the toxicity targets of cadmium were screened. It was proved that chlorophyll a, photosynthetic electron transport chains and ribosomes were confirmed as the toxicity targets, and the resistance mechanism of cadmium was the up-regulation of ABC transporter pathway protein expression. This manuscript is very interesting and the novel finding on cadmium stress in photosynthetic organisms was disclosed. In my opinion, this manuscript could be accepted after minor revision.
1.The tense of some sentences in the full text is problematic and needs to be modified.
2. In the abstract section, “the mechanism of cadmium resistance was the up-regulation of ABC transporter pathway protein expression” is different from the conclusion “ the overexpression of cation efflux system proteins and ABC transporters on the cell membrane proved that ion efflux is the principal resistance mechanism of Synechocystis sp. to cadmium stress”, please check and modify it.

Reviewer 3 Report

Comments:

The manuscript entitled “Molecular Responses Mechanism of Synechocystis sp. PCC 6803 to Cadmium Stresshas many grammatical issues that need to be corrected. The authors need to proofread the manuscript thoroughly. I suggest the following changes and improvements:

1.     Authors need to revise the abstract section considering the important findings and underscore the scientific value added to your paper in your abstract.

2.     The current structure of the introduction is not well organized and well written. The authors need to be improved.

3.     Avoid redundant words at the start of the sentence such as, But, Also, etc.

4.     What are the current research gap and the significance of this work?

5.     The novelty of this work should be stated clearly in the introduction section.

6.     For the consistency of the manuscript, the word should be consistent throughout the text according to journal guidelines, such as (minutes or min), (hour or h), (Figure or Fig.), (ml or mL), etc.

Reviewer 4 Report

The manuscript studied the effect of Cadmium on Synechocystis sp. PCC 6803 and the molecular responses were investigated. The paper is well written. Only a few questions need to be solved.

-Why do the authors choose the Cadmium concentration range of 0 to 1 mg/L? What’s the application situation? More detailed information should be added in the introduction part.

- Don’t use acronyms when they appear first. Please check thoroughly.
